# A High Polyphenol Diet Improves Psychological Well-Being: The Polyphenol Intervention Trial (PPhIT)

**DOI:** 10.3390/nu12082445

**Published:** 2020-08-14

**Authors:** Meropi D. Kontogianni, Aswathy Vijayakumar, Ciara Rooney, Rebecca L. Noad, Katherine M. Appleton, Danielle McCarthy, Michael Donnelly, Ian S. Young, Michelle C. McKinley, Pascal P. McKeown, Jayne V. Woodside

**Affiliations:** 1Department of Nutrition and Dietetics, Harokopio University, Eleftheriou Venizelou 70, 17671 Kallithea, Greece; mkont@hua.gr; 2Centre for Public Health, Queen’s University Belfast, Belfast BT12 6BA, UK; a.vijayakumar@qub.ac.uk (A.V.); crooney34@hotmail.com (C.R.); rebeccanoad@gmail.com (R.L.N.); michael.donnelly@qub.ac.uk (M.D.); I.Young@qub.ac.uk (I.S.Y.); m.mckinley@qub.ac.uk (M.C.M.); P.P.McKeown@qub.ac.uk (P.P.M.); 3Cardiology Department, Belfast Health and Social Care Trust, Belfast BT9 7AB, UK; 4Department of Psychology, Bournemouth University, Bournemouth BH12 5BB, UK; k.appleton@bournemouth.ac.uk; 5Institute for Global Food Security, Queen’s University Belfast, Belfast BT9 5DL, UK; D.McCarthy@qub.ac.uk

**Keywords:** polyphenols, fruits, berries, vegetables, dark chocolate, psychological well-being, depression, physical health, mental health

## Abstract

Mental ill health is currently one of the leading causes of disease burden worldwide. A growing body of data has emerged supporting the role of diet, especially polyphenols, which have anxiolytic and antidepressant-like properties. The aim of the present study was to assess the effect of a high polyphenol diet (HPD) compared to a low polyphenol diet (LPD) on aspects of psychological well-being in the Polyphenol Intervention Trial (PPhIT). Ninety-nine mildly hypertensive participants aged 40–65 years were enrolled in a four-week LPD washout period and then randomised to either an LPD or an HPD for eight weeks. Both at baseline and the end of intervention, participants’ lifestyle and psychological well-being were assessed. The participants in the HPD group reported a decrease in depressive symptoms, as assessed by the Beck Depression Inventory-II, and an improvement in physical component and mental health component scores as assessed with 36-Item Short Form Survey. No differences in anxiety, stress, self-esteem or body image perception were observed. In summary, the study findings suggest that the adoption of a polyphenol-rich diet could potentially lead to beneficial effects including a reduction in depressive symptoms and improvements in general mental health status and physical health in hypertensive participants.

## 1. Introduction

Mental ill health, manifesting itself in a wide range of conditions such as depression, anxiety and stress [1], represents one of the leading causes of burden of disease worldwide, also substantially increasing the risk of cardiovascular disease, diabetes and cancer [2,3,4] and adversely affecting quality of life (QoL), relationships and the ability to work [5]. Northern Ireland has the highest prevalence of mental illness within the UK, and psychiatric morbidity is 25% higher than in the UK [6].

Thus, research is required in order to establish inexpensive and effective techniques to reduce the incidence of mental health problems and to improve the psychological well-being of the population. Alongside genetic and biological factors, researchers have increasingly begun to examine the role of lifestyle factors, including dietary intake, in the promotion of psychological well-being and the prevention of mental illness [7,8]. Studies that have explored potential associations between nutrient intake (namely carbohydrates, B vitamins and antioxidants such as vitamins C, E and polyphenols) or foods rich in these nutrients (e.g., fruits, vegetables, legumes, coffee, chocolate) and psychological well-being have produced conflicting results [9,10,11,12].

Polyphenols, in particular, have gained increasing attention from health researchers in recent years due to their biological properties, as well as their abundance within the human diet [13]. A growing number of epidemiological studies support a role for polyphenols in the prevention of chronic non-communicable diseases such as cardiovascular disease (CVD) [14], cancer [15] and neurodegenerative diseases [14,16]. Furthermore, animal studies have demonstrated the ability of polyphenols to improve cognitive performance and memory [17,18] and, more recently, these results have been replicated in human studies [19,20]. Regarding mental health, a growing body of data from animal and human studies has emerged supporting the role of a variety of dietary polyphenols in affecting behaviour and mood through anxiolytic and antidepressant-like properties, mediated through multiple molecular and cellular pathways [21]. Moreover, given that recent studies have demonstrated the pathophysiological role of oxidative stress and inflammation in the onset and progression of depression, polyphenols have been examined both in vitro and in vivo as a potential antidepressant treatment, although randomised controlled trials are still scarce in the field [22,23]. The richest sources of polyphenols in the human diet include fruits (e.g., berries, grapes, apples and plums), vegetables (e.g., cabbage, eggplant, onions, peppers), plant-derived beverages including tea, coffee, red wine and fruit juices (e.g., apple juice), seeds, nuts and chocolate (particularly dark chocolate) [24,25]. In terms of a food-based approach, several of the above-mentioned foods have been studied both in observational and intervention studies for potential effects on outcomes related to mental well-being, mood, psychological distress and life satisfaction [26], although, potentially due in part to the great variation in study design, results are not consistent. Studying diet on a dietary pattern level will be beneficial in allowing potential complicated or cumulative intercorrelations, interactions and synergies to be revealed, given that different polyphenols may have different effects on outcomes of mental health [27,28,29].

The aim of the present study was to assess the effect of a polyphenol-rich dietary pattern (comprising fruits, including berries; vegetables and dark chocolate) in comparison to a control diet (low fruits and vegetables, <2 portions/day, and no dark chocolate) on aspects of psychological well-being and mental health status including mood, QoL, body image perception and self-esteem as secondary outcomes measured within the Polyphenol Intervention Trial (PPhIT) [30].

## 2. Materials and Methods 

### 2.1. Setting and Study Population

PPhIT was a randomised, controlled, parallel-group, single-blinded dietary intervention trial, primarily designed to test whether increasing overall polyphenol dietary intake would affect microvascular function and a range of other markers of CVD risk, such as systolic blood pressure and lipid profile, in patients with hypertension. All participants underwent a full assessment at baseline (week 0) (described below); then, they entered a washout period, during which they consumed a low polyphenol diet, and afterwards were randomised to either a low polyphenol diet (LPD) or a high polyphenol diet (HPD) group for 8 weeks (Figure 1). A full assessment was repeated for all the participants at the end of the 8-week intervention (week 12), while at the end of the washout period (week 4), participants also underwent a dietary intake assessment, anthropometric measurements and blood and urine sample collection.

Participants aged 40–65 years, with documented grade I (140–159/90–99 mmHg) or grade II (160–179/100–109 mmHg) hypertension, were eligible. Participants with diabetes mellitus, acute coronary syndrome or transient ischaemic attack within three months, pregnancy or lactation, fasting triglyceride concentration >4 mmol/L, alcohol consumption (>28 units/week for men and >21 units/week for women), oral anticoagulant therapy or antioxidant supplements, dietary restrictions that would limit ability to comply with the study diets, body mass index >35 kg/m^2^ or with an impalpable brachial artery were excluded from the study. Recruitment for PPhIT began in February 2011 and was completed by January 2013. All participants were informed about the aims and procedures of the study and gave their written consent. The study had ethical approval from the Office of Research Ethics Committee Northern Ireland (ref 10/NIR03/39) and was registered at ClinicalTrials.gov (ref NCT01319786). Details of the primary aim of the study, population, design, recruitment procedures and main findings have been published elsewhere [30]. Below, we provide some additional details on selected aspects of the evaluation that pertain to the analyses reported in this manuscript. 

### 2.2. Dietary Intervention

The intervention commenced with a four-week “washout period” for all participants, during which they were asked to consume two portions or less of fruits and vegetables (F&V) per day and to exclude berries and dark chocolate (LPD). At the end of this period, subjects were randomised to either continue with the above LPD for a further 8-week “intervention period” or to consume an HPD of six portions of F&V (including one portion of berries per day) and 50 g of dark chocolate per day (Figure 1). A portion of fruit and vegetables was quantitatively defined using household measures as outlined by UK guidelines (https://www.nhs.uk/live-well/eat-well/5-a-day-portion-sizes/), i.e., 1 apple, 1 orange, half a grapefruit or one glass (150 mL) fruit juice, 3 tablespoons of vegetables [31]. All participants in the HPD had a self-selected weekly delivery of F&V and dark chocolate (Lindt^®^ 70% cocoa) free of charge to their homes from a local supermarket and were provided with written material regarding F&V portion sizes, recipes and sample diet plans. In addition, each participant, regardless of dietary allocation, was also contacted by telephone at weekly intervals to provide support and encouragement and to discuss potential barriers encountered in relation to achieving the dietary goals.

Dietary intake and compliance with the intervention were assessed through 4-day food diaries completed on four occasions: on the four days leading up to the week 0 visit (baseline measurement), on the four days leading up to the week 4 visit (washout period measurement), at week 8 (intervention measurement) and on the four days leading up to the final week 12 visit (a second intervention period measurement). Circulating blood and urine levels of a panel of nutritional biomarkers, with detailed methodology given below, were also used to assess compliance. Self-reported F&V, berries and dark chocolate consumed per day (as recorded in the food diaries) were extracted and entered into a Microsoft Excel spreadsheet. The spreadsheet contained pre-determined formulae which transformed the actual amounts of F&V and berries consumed into “portions” according to the “5-a-day” message. 

### 2.3. Other Lifestyle Parameters

A “lifestyle and medical” questionnaire was used at week 0 to record participant demographic, lifestyle and medical information. The questionnaire had 16 items in total and assessed several aspects including vitamin and mineral supplement usage, smoking and alcohol habits, history of education, current occupational status, current medication, history of steroid use and, for females, use of hormone replacement therapy and details of menstrual cycle. Information regarding changes in medication use, smoking and alcohol patterns, as well as infections/illnesses were also recorded throughout the study.

Participants’ physical activity levels were recorded at weeks 0, 4 and 12 to ensure that habitual activity levels were not altered for the duration of the study. The Recent Physical Activity Questionnaire (RPAQ), designed by the Medical Research Council (MRC Epidemiology Unit, Cambridge, UK), was used to measure physical activity. The questionnaire assesses physical activity within the preceding four weeks based on three primary areas: activity at home, activity at work (including travel to and from work) and recreational activities. The RPAQ has been shown to be a valid instrument for calculating total energy expenditure, physical activity energy expenditure and physical activity in healthy adults [32]. In terms of analysis, physical activity as recorded in the questionnaire was converted to total metabolic equivalent of task (MET) hours per day of sedentary, light, moderate and vigorous activity.

### 2.4. Anthropometric, Clinical and Biochemical Assessments

Participants attended the Royal Victoria Hospital (Belfast, Northern Ireland, UK) for assessments on three occasions throughout the study: baseline (week 0), washout period (week 4) and intervention (week 12). Body weight of participants was measured to the nearest 100 g and height to the nearest 0.5 cm. Body mass index (BMI) was calculated as weight (kg) divided by height squared (m^2^). Waist circumference (WC) and hip circumference (HC) were tape-measured to the nearest 0.1 cm. Blood pressure (BP) was measured using an Omron M5-1 automatic BP monitor (Omron Healthcare, Hoofddorp, The Netherlands). Three consecutive readings were recorded, and a mean BP was calculated from the 2nd and 3rd readings. To measure the primary endpoint of PPhIT (microvascular function), venous occlusion plethysmography was conducted on participants by determining forearm blood flow during incremental intra-arterial infusions of acetylcholine and sodium nitroprusside, as previously described [30]. Blood samples were also collected. Fasting serum lipid profiles (total cholesterol, high density lipoprotein (HDL) and triglycerides) were assessed using standard enzymatic colorimetric assays on an automated Cobas^®^ 8000 Modular system biochemical analyser (Roche Diagnostics Ltd., West Sussex, UK). Low-density lipoprotein (LDL) cholesterol was calculated using a standard Friedewald formula [33]. Blood and urine markers of micronutrient status were assessed at weeks 0 and 12 to objectively measure compliance to the intervention diet. Plasma vitamin C was measured on a FLUOstar Optima plate reader (BMG Labtech, Ortenberg, Germany), adapted from the method of Vuilleumier and Keck [34]. Serum concentrations of six carotenoids (α-carotene, β-carotene, β-cryptoxanthin, lutein, lycopene and zeaxanthin) were measured by reverse phase high performance liquid chromatography (HPLC), as described by Craft [35]. Urine collected from the volunteers between evening meal and midnight the evening before each study visit was analysed, including an enzymatic hydrolysis step, to quantify total epicatechin content, using an Agilent Technologies 1100 series HPLC (Agilent Technologies, Stockport, UK) directly linked to a Waters Micromass Quattro Ultima Platinum API triple quadrupole mass spectrometer (Waters, Dublin, Ireland).

### 2.5. Psychological Well-Being, Self-Esteem and Body Image

Aspects of psychological well-being and mental health status were evaluated through several scales and questionnaires that were completed at weeks 0 and 12. The decision to use these questionnaires only twice was made for three reasons: (i) the study wished to investigate the effect of the intervention diet in comparison to normal psychological well-being, rather than psychological well-being under the controlled conditions of the washout period; (ii) to reduce participants’ burden at week 4 visits, which were already long (2.5 h) in duration due to vascular function and dietary assessments; (iii) distributing the surveys at three time points may have been disadvantageous in terms of allowing participants to become familiar with their format, which may have influenced responses. All questionnaires are commonly used for assessing various aspects of mental health and psychological well-being in the general population. 

The Positive and Negative Affect Schedule (PANAS) was used for evaluating subjective mood. The questionnaire measures two distinctive dimensions: positive affect (PA) and negative affect (NA) [36]. PA is associated with pleasurable engagement with the environment, including feelings of enthusiasm and alertness as well as feeling active. NA refers to unpleasurable engagement with the environment, comprising feelings of anger, contempt, disgust, guilt, fear and nervousness. Whilst related, PA and NA represent two distinct and independent dimensions of mood. Participants were asked to respond to 10 items representing PA and 10 items representing NA on a five-point scale. Higher scores represent higher positive and negative affect, respectively. This was the only questionnaire assessing psychological well-being that was also completed at week 4, in order to monitor psychological well-being at the end of the washout period. 

Depressive symptomology was assessed with the Beck Depression Inventory-II (BDI-II), a 21-item, self-report questionnaire developed by Beck and colleagues [37]. Each item on the BDI-II has four statements which relate to the severity of a particular depressive symptom, and respondents are asked to choose the one statement which best describes how they have been feeling in the preceding two weeks. Higher scores indicate higher levels of depression (scores 0–13 = minimal; 14–19 = mild; 20–28 = moderate; 29–63 = severe). The shorter version (21 items) of the Depression Anxiety Stress Scale (DASS-21) was also completed to measure depression, anxiety and stress [38]. The DASS-21 questionnaire was introduced nine months into the recruitment of the participants. DASS-21 has seven items per subscale and asks participants to rate the extent to which they experienced each emotional state the preceding week using a four-point Likert scale (0 = Did not apply to me at all, 3 = Applied to me very much or most of the time). Higher scores are indicative of higher levels of depression, anxiety and stress. 

The Rosenberg Self-Esteem Scale was used as a global measure of self-esteem [39,40]. The questionnaire consists of a ten-item Likert scale, completed using a four-point scale from strongly agree to strongly disagree. Scores can range from 0 to 30, with higher scores indicating higher self-esteem. Finally, body image satisfaction was assessed through the Multidimensional Body Self-Relations Questionnaire—Appearance Scales (MSRQ-AS), a 34-item validated measure of body image perception for use in general populations (www.body-images.com) [41]. This version contains five subscales: appearance evaluation (satisfaction with ones looks), appearance orientation (levels of investment in one’s appearance), overweight preoccupation (weight anxiety, vigilance, dieting etc), self-classified weight (how one perceives and labels one’s weight) and body area satisfaction (satisfaction with areas of body). The questionnaire contains a series of statements and asks participants to indicate the extent to which each statement applies to them personally, with higher scores generally indicating greater body image satisfaction.

### 2.6. Mental and Physical Health

Mental and physical health were assessed with the RAND Medical Outcomes Study 36-item Short Form Health Survey (SF-36) [42,43]. A total of 36 questions are included in the RAND SF-36 survey and eight key areas are explored in the SF-36 including physical functioning, role limitations due to physical problems, pain, general health, energy/fatigue, social functioning, role limitations due to emotional problems and emotional well-being. The raw data were recoded using the RAND SF-36 scoring instructions available online. Additionally, the eight areas were combined to obtain the scoring for physical and mental health components. As the eight different components consist of different numbers of questions, the normal scores were transformed to T-scores, as described by Hays et al. 1993 [44] and Hays et al. 1995 [45]. Physical functioning, role limitations due to physical problems, pain and general health were combined to obtain the physical health component and role limitations due to emotional problems, energy/fatigue, emotional well-being and social functioning were combined to obtain the mental health component.

### 2.7. Statistical Analyses

The sample size calculation was based on the PPhIT primary outcome, namely microvascular function. According to this, detection of a 33% difference between groups in microvascular function, measured by forearm blood flow responses to an endothelium-dependent vasodilator, with 90% power, using a 2-tailed test at the 5% significance level, would require 50 participants per group. The current analysis reports secondary outcomes, for which power calculations were not performed.

Results are expressed as mean ± standard deviation for normally distributed continuous variables and as medians and interquartile ranges for continuous skewed variables. Categorical variables are presented as absolute (*n*) and relative frequencies (%). The normality of variables was checked through the Shapiro–Wilk test and graphically through histograms. Concentration measures of micronutrients were logarithmically transformed and were summarised as geometric median and interquartile range. The principal analysis for each outcome variable was a between-group comparison of change using independent sample t tests or Mann–Whitney U test for continuous parametric and non-parametric variables, respectively, and chi-squared test for categorical variables. Within-group comparisons were performed using paired sample t tests or Wilcoxon signed-rank test for parametric and non-parametric continuous variables, respectively. Statistical significance was set at *p* ≤ 0.05. All statistical methods were conducted using PASW Statistics 18 for Windows (SPSS Inc., Chicago, IL, USA).

## 3. Results

### 3.1. General Results

Ninety-nine participants completed the PPhIT study, including 53 (53.5%) males. Participants had a mean age of 54.9 ± 6.9 years, with ages ranging from 40 to 65 years. The majority (52%) of the sample were obese (BMI ≥ 30 kg/m^2^). In total, 12.1% were current smokers, and 43.4% stated that they had smoked in the past. Baseline characteristics according to dietary group (LPD versus HPD) are shown in Table 1. Overall, the groups were similar upon entering the study, with no statistically significant differences in anthropometric, lifestyle and basic clinical characteristics.

During the washout period, no changes were recorded in participants’ physical activity habits, weight status, smoking habits, medication use or clinical condition compared to baseline in both HPD and LPD groups (data not shown). Additionally, mood evaluation according to PANAS questionnaire did not record any change between baseline and end of washout period (both *p* > 0.05) (data not shown). F&V intake per day declined significantly during washout, from 2.67 portions at week 0 to 1.38 portions at week 4 within the overall sample (*p* < 0.001), and significant reductions in blood levels of vitamin C (*p* < 0.001) and β-cryptoxanthin (*p* = 0.05), but not in any of the other carotenoids measured, were also recorded (data not shown).

Dietary intake of food groups and micronutrients, as well as weight status and physical activity levels both at baseline and at the end of the intervention period, are presented in Table 2, per intervention group. At baseline, there was no significant difference in intake of F&V, berries and dark chocolate and concentration of micronutrients between the LPD and HPD group. By the end of the intervention, there was a significant increase in intake of F&V, berries and dark chocolate in the HPD group, and the differences in change in intake between the two groups were statistically significant. Furthermore, there was a significant increase in the concentration of biomarkers, plasma vitamin C, serum lutein, β-cryptoxanthin, α-carotene and lycopene and urinary epicatechin over the course of the intervention in the HPD group, and the differences in the change in the concentration between the LPD and HPD group were statistically significant. These results indicate good compliance with the intervention diet, with significant between-group differences in change in all biomarkers measured except β-carotene. No differences were recorded in change in physical activity and weight status between the two intervention groups during the intervention.

### 3.2. Changes in Aspects of Psychological Well-Being

Changes in measures of psychological well-being between baseline and intervention are illustrated in Table 3. There were no significant differences in scores of BDI-II, DASS-21 or PANAS between the LPD and HPD groups at baseline. There was a significant between-group difference (*p* = 0.01) in change in depressive symptoms as assessed with BDI-II, but no other significant effects were found between groups with regards to depression, anxiety or stress measured using the DASS-21 or positive and negative affect measured with PANAS. Regarding within-group changes, a borderline significant (*p* = 0.05) result was detected for a reduction in stress measured by DASS-21 within the HPD group, as well as an improvement in subjective mood (positive affect) (*p* = 0.03) measured by PANAS.

### 3.3. Changes in Self-Esteem and Body Image Perception

There were no significant differences in self-esteem or body image perception scores between the LPD and HPD groups at baseline. As shown in Table 3, there were also no significant differences between the HPD and LPD in self-esteem or body image perception scores at the end of the intervention.

### 3.4. Changes in Health-Related Quality of Life

There were no significant differences between groups at baseline with regards to health-related quality of life measured using the SF-36. There were statistically significant between-group differences in change in different component scores (general health (*p* = 0.03) and energy/fatigue (*p* = 0.02)) and the overall summary scores for the physical health component (*p* = 0.04) and mental health component (*p* = 0.01), with more positive changes demonstrated in the HPD group. In the HPD group, there were also within-group improvements in role limitations due to physical health (*p* = 0.04), general health (*p* = 0.00), energy/fatigue (*p* = < 0.001), emotional well-being (*p* = < 0.001) and social functioning (*p* = 0.02)

## 4. Discussion

Given the high prevalence of mental health problems and the potential effect of dietary patterns on their onset and/or treatment, the aim of the present study was to assess the effect of a polyphenol-rich dietary pattern (comprising fruits, including berries; vegetables and dark chocolate) on aspects of psychological well-being or mental health status, including mood, self-esteem and body image perception, as secondary outcomes of the PPhIT study. Despite some heterogeneity, the study findings suggest that the adoption of such a polyphenol-rich diet could potentially lead to beneficial effects on certain outcomes including depressed mood and physical and mental health in hypertensive participants.

There was a significant difference in change in depressive symptoms assessed with BDI-II between the HPD group and the LPD group, indicating a positive effect of the HPD, which is in agreement with a number of other observational studies focusing on the same outcome measure and polyphenol-rich foods. In the HPD group, a 66.6% reduction in BDI-II score was observed after the intervention. Button et al. 2015, using data from three randomised controlled trials (RCT) with a sample of *n* = 1039, identified that a 17.5% reduction in score was necessary to observe minimally clinically important differences [46]. Oliveria et al. (2019) found a negative association between depressive symptoms measured by BDI and high intake of polyphenol food items [47]. In the Finnish general population (*n* = 2011), daily intake of tea was associated with reduced risk of depressive symptoms defined by BDI scores [48]. Similarly, in the Mediterranean healthy eating, lifestyle and aging (MEAL) study (*n* = 1572), the dietary intake of phenolic acid, flavanones and anthocyanin were negatively associated with depressive symptoms measured using the Center for Epidemiologic Studies Depression Scale (CES-D-10) [49]. The positive effects observed in the present study may be attributable to other nutrients found in F&V, berries and dark chocolate which may work independently or synergistically to influence health outcomes. Brody (2002) found that vitamin C intake over a 14 day period led to a moderate reduction in depressive symptoms amongst 42 healthy young adults [50]. In our study, there was a significant difference in plasma vitamin C status between the LPD and HPD group. Similarly, there were significant increases in serum carotenoids, lutein, zeaxanthin, β-cryptoxanthin and urinary epicatechin within the HPD group, and some studies have suggested a link between these nutrients and improvements in psychological well-being including depression [51]. The antidepressant effects of polyphenols may be associated with both their antioxidant and anti-inflammatory properties, whereby there is a reduction in free radicals and cytokine dysregulation [12]. Lua and Wong (2012) found that the consumption of 50 g dark chocolate (70% cocoa) for three days was associated with significant improvement in depressed mood [52].

The primary outcome of the PPhIT study was to identify whether high consumption of F&V, berries and dark chocolate could improve microvascular function in hypertensive subjects [30]. High intake of polyphenol, specifically including F&V, berries and dark chocolate in diet, resulted in significant improvements in endothelium-dependent (acetylcholine) vasodilator [30]. Depression is often observed among individuals with vascular diseases such as hypertension, peripheral vascular disease and coronary artery disease, known as “vascular depression hypothesis” [53]. Studies have reported morphological changes in vascular structure and altered expression of endothelial cell molecules such as nitric oxide in patients with depression [53]. In the current study, the improvements in endothelium-dependent vasodilatation might have also resulted in improvements in depressed mood.

In light of the findings from the BDI-II, it is interesting that no notable effects of the polyphenol-rich diet were observed on depressed mood measured using the DASS-21 questionnaire in this study. The DASS-21 questionnaire was introduced as an amendment to PPhIT, given concerns that BDI-II is used to screen for depression in normal populations or to assess severity of depression in clinical populations, and therefore it was thought possible by the research team that the tool may not have been sensitive enough to pick up changes due to diet. Page et al. (2007) showed that DASS-21 has good psychometric properties and is moderately sensitive to changes that result from the treatment [54]. However, this resulted in a considerably smaller sample size for the analysis of the DASS-21 questionnaire (*n* = 57) compared to BDI-II, which may have had implications in terms of the associated power available to detect differences between the two diet groups. The DASS-21 also showed no statistically significant differences in change between groups with regards to stress or anxiety. Furthermore, for both measures, scores on all scales at the start of the study are low, and negative affect scores for the PANAS are also low. These low scores are unsurprising in a volunteer sample for a study intended to improve health but may also have limited our chances of finding effects. Further study in groups with higher levels of poor psychological health, e.g., those with diagnoses of depression or anxiety, may be of value.

In the present study, significant improvements in quality of life between the HPD group and the LPD group measured using the SF-36 health survey questionnaire were found. There were statistically significant improvements in both physical and mental health components in the HPD group when compared with the LPD group. Data showing the effect of dietary interventions and especially of polyphenols/antioxidants on quality of life parameters are sparse and mainly limited to patients with chronic diseases such as multiple sclerosis, chronic fatigue syndrome and depression. Steptoe et al. (2004) found that a higher intake of fruits and vegetables through behavioural and nutrition education counselling was positively associated with physical health status but not mental health status measured using SF-36 among adults in a low-income neighbourhood [55]. A sub-study of the DASH trial also found that adhering to a fruit and vegetable-rich diet was associated with improved perception of quality of life [56]. It is important to acknowledge that while the self-reported improvements in physical and mental health scores observed within the current sample may be attributed to the foods consumed, they may also be wholly or partly influenced by taking part in the intervention and increased positivity that may come from making positive dietary changes. As pointed out in the study by Plaisted et al. (1999), improvements in QoL might be attributable to participants’ awareness that they are consuming a healthy diet, which could have contributed to improved self-ratings of general health and mental health component [56]. In addition, given that depressive symptoms were improved in the HPD group, the improvements in mental health component may simply mirror these findings.

It is important to consider the results of this study in light of a number of methodological limitations. Firstly, as the primary purpose of PPhIT was to test the effect of a polyphenol-rich dietary pattern on microvascular and platelet function, the outcomes discussed here are secondary endpoints. Hence, as mentioned previously, it is possible that the study may not have been adequately powered to detect differences between the dietary groups, which may account for some of the null findings demonstrated. Secondly, the study sample comprised mildly hypertensive participants, which limits the generalisability of these results to the wider population. Furthermore, it is possible that selection bias exists within the current sample, given that the volunteers for this study were on the whole well-educated, and, as is the case with most clinical trials, are likely to have been more motivated with regard to improving their health than the general population. The participants in the HPD group were provided with the key intervention foods on a weekly basis, whereas the LPD group received no food provision as their diet was to remain unchanged. This may have increased the likely compliance of the HPD group with the intervention. Another limitation of this study was the use of self-report measures to measure psychological outcomes. Self-report measures can be disadvantageous in that they can be affected by forms of bias, including response, recall and social desirability bias, which can lead to inaccurate responses and conclusions [57]. However, given the subjective nature of psychological well-being, self-reporting is the most suitable method of obtaining information on individuals’ personal experiences and emotions. The current study employed validated and previously used measures to collect information on individuals’ personal experiences and emotions [58,59]. Additionally, it must be noted that the questionnaires described in this study were distributed at week 0 (baseline) and week 12 (intervention). It is possible that the washout period (week 0 to week 4) could have potentially affected people’s psychological well-being and thus it may have been useful to additionally measure the endpoints at week 4. However, the decision to distribute the questionnaires at week 0 and week 12 was made for three main reasons: (i) to reduce participant burden at week 4 visits, which were already long (2.5 h) in duration due to vascular function assessment and the dissemination of dietary advice; (ii) the study wished to investigate the effect of the intervention diet in comparison to normal psychological status, rather than psychological states under controlled conditions, which would have limited the applicability of the results; (iii) distributing the outcome measures at three time points may have been disadvantageous in terms of allowing participants to become familiar with their format, which may have induced response bias. Another limitation common to most studies analysing self-reported questionnaire data is the number of variables assessed, which may have increased the chance of type one errors (identification of the false positive) associated with hypothesis testing.

In contrast, one of the most obvious strengths of this study is its RCT study design. However, as the randomisation according to the groups only occurred at week 4, the presentation of week 0 data based on the allocated groups is rather artificial, and this must be considered a limitation. As further strengths, the study implemented a variety of techniques to encourage and monitor compliance. As a result of such efforts, participants were demonstrated to have good compliance with both diets, which was assessed both subjectively and objectively. Furthermore, the study had good retention of participants, with a less than 5% (*n* = 5) drop out level, all of which were due to reasons unrelated to the study.

## 5. Conclusions

In conclusion, the results from the present RCT trial showed heterogeneous findings regarding the effect of a polyphenol-rich dietary pattern on aspects of psychological well-being, with positive effects demonstrated on depressive symptoms and both the physical and mental health status components of the SF-36 quality of life measure. Further studies with psychological well-being impacts as primary endpoints, with appropriate study design and sample sizes, are needed in order to confirm the benefits of a polyphenol-rich dietary pattern on these outcomes.

## Figures and Tables

**Figure 1 nutrients-12-02445-f001:**
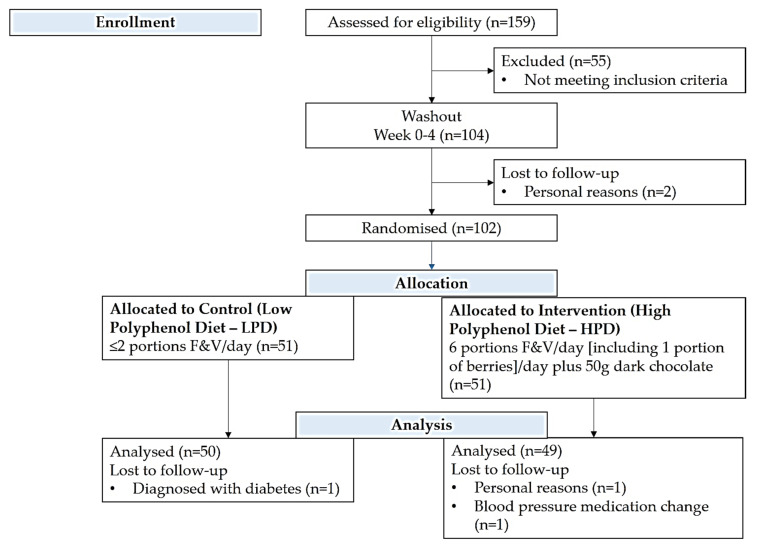
CONSORT diagram summarizing flow of participants through the study.

**Table 1 nutrients-12-02445-t001:** Baseline (week 0) participant characteristics according to the Polyphenol Intervention Trial (PPhIT) study group allocation.

	Low Polyphenol (*n*_max_ = 50)	High Polyphenol (*n*_max_ = 49)	Between-Group Comparison *p*-Value *
Age (years)	55.6 ± 6.8	54.0 ± 7.0	0.25
Sex (males, *n* (%))	30 (60.0)	23 (46.9)	0.23
Education (years)	13.9 (12.0, 16.8)	13.6 (13.8, 15.8)	0.57
Body mass index (kg/m^2^)	29.9 (26.9, 34.6)	31.15 (27.7, 33.5)	0.29
Waist circumference (cm)			
*Male*	106.5 (98.0, 116.3)	105.0 (98.0, 116.0)	0.86
*Female*	94.0 (85.3, 108.8)	96.0 (89.5, 108.5)	0.56
Current smoker *n* (%)	5 (10.0)	7 (14.3)	0.55
Use of antidepressants *n* (%)	7 (14.0)	8 (16.3)	0.79
Systolic blood pressure (mmHg)	143.7 ± 6.6	143.6 ± 8.0	0.95
Diastolic blood pressure (mmHg)	86.9 ± 8.3	85.9 ± 7.1	0.55
Total cholesterol (mmol/L)	5.3 ± 1.2	5.7 ± 1.2	0.10
HDL(mmol/L)	1.4 ± 0.3	1.3 ± 0.4	0.58
LDL(mmol/L)	4.3 ± 1.3	4.6 ± 1.2	0.18
Triglycerides (mmol/L)	1.7 (1.6, 2.1)	1.6 (1.5, 2.0)	0.46

HDL—high-density lipoprotein; LDL—low-density lipoprotein. Continuous variables are summarised as mean ± SD or medians and interquartile ranges. Categorical variables are summarised as *n* (%). * Between-group comparisons were made using independent sample t-tests (*p* < 0.05) or Mann–Whitney U test for continuous variables and chi-squared tests (*p* < 0.05) for categorical variables.

**Table 2 nutrients-12-02445-t002:** Baseline (week 0) and end of intervention (week 12) dietary intake, plasma, serum and urine micronutrient biomarkers, physical activity and weight status characteristics according to the Polyphenol Intervention Trial (PPhIT) study group allocation.

	Low Polyphenol Diet (*n*_max_ = 50)				High Polyphenol Diet (*n*_max_ = 49)				
	Week 0 ^1^	Week 12	Median Change (IQR) ^3^	Within Group (*p* Value) ^4^	Week 0 ^1^	Week 12	Median Change (IQR) ^3^	Within Group (*p* Value) ^4^	Between Group (*p* Value) ^5^
***Dietary intake***									
Fruits and vegetables intake (portions/day)	2.68 ± 1.68)	1.24 ± 0.56	−1.44 (−1.87, −0.95)	**<0.001**	2.64 ± 1.70	6.73 ± 2.07	4.09 (3.45, 4.73)	**<0.001**	**<0.001**
	**Week 0 ^1^**	**Week 12**	**Median Change (IQR) ^3^**	**Within Group (*p* value) ^4^**	**Week 0 ^1^**	**Week 12**	**Median Change (IQR) ^3^**	**Within Group (*p* value) ^4^**	**Between Group (*p* value) ^5^**
Berries (portions/day)	0 (0.00, 0.00)	0 (0.00, 0.00)	0 (0.00, 0.00)	0.69	0 (0.00, 0.00)	1 (0.80, 1.25)	1 (0.75, 1.17)	**<0.001**	**<0.001**
Dark chocolate (grams/day)	0 (0.00, 0.00)	0 (0.00, 0.00)	0 (0.00, 0.00)	0.18	0 (0.00, 0.00)	50 (37.50, 50.00)	50 (37.5, 50.0)	**<0.001**	**<0.001**
***Micronutrient biomarkers***									
Vitamin C (µmol/l) ^6^	44.3 (28.70, 61.90)	34.2 (13.40, 49.50)	−7.60 (−26.50, 0.85)	**<0.001**	46.4 (31.70, 65.00)	55.7 (43.10, 68.20)	4.83 (−8.68, 20.75)	0.1	**<0.001**
Total carotenoids (µmol/l) ^6^	1.09 (0.88, 1.43)	1.09 (0.72, 1.35)	−0.01 (−0.03, 0.01)	**0.01**	1.17 (0.96, 1.48)	1.33 (1.06, 1.66)	0.12 (−0.12, 0.40)	0.07	**<0.001**
Lutein (µmol/l) ^6^	0.15 (0.12, 0.22)	0.14 (0.11, 0.20)	−0.01 (−0.03, 0.01)	0.08	0.14 (0.11, 0.20)	0.20 (0.15, 0.26)	0.04 (0.01, 0.07)	**<0.001**	**<0.001**
Zeaxanthin (µmol/l) ^6^	0.04 (0.03, 0.05)	0.03 (0.03, 0.05)	−0.00 (−0.01, 0.01)	0.3	0.04 (0.03, 0.05)	0.04 (0.04, 0.06)	0 (−0.00, 0.01)	**0.01**	**0.01**
β-cryptoxanthin (µmol/l) ^6^	0.06 (0.04, 0.07)	0.05 (0.03, 0.08)	−0.01 (−0.02, 0.01)	**0.049**	0.06 (0.04, 0.09)	0.07 (0.05, 0.09)	0.01 (−0.01, 0.03)	**0.03**	**<0.001**
α-carotene (µmol/l) ^6^	0.12 (0.09, 0.15)	0.1 (0.08, 0.15)	−0.01 (−0.04, 0.01)	0.06	0.13 (0.11, 0.18)	0.15 (0.11, 0.20)	0.01 (−0.02, 0.03)	0.17	**0.02**
β-carotene (µmol/l) ^6^	0.23 (0.16, 0.34)	0.22 (0.13, 0.32)	−0.01 (−0.09, 0.03)	0.07	0.27 (0.19, 0.44)	0.3 (0.21, 0.41)	−0.00 (−0.06, 0.08)	0.79	0.14
Lycopene (µmol/l) ^6^	0.49 (0.38, 0.66)	0.47 (0.33, 0.58)	−0.04 (−0.19, 0.09)	0.07	0.5 (0.39, 0.61)	0.56 (0.40, 0.68)	0.05 (−0.14, 0.21)	0.31	**0.048**
Epicatechin (nmol/mg crt/L) ^6^	0.43 (0.17, 1.10)	0.55 (0.26, 1.00)	0.04 (−0.24, 0.32)	0.37	0.64 (0.25, 0.97)	1.73 (0.51, 4.20)	0.89 (0.10, 3.57)	**<0.001**	**<0.001**
	**Week 0 ^1^**	**Week 12**	**Mean Change (95% CI) ^2^**	**Within Group (*p* value) ^4^**	**Week 0 ^1^**	**Week 12**	**Mean Change (95% CI) ^2^**	**Within Group (*p* value) ^4^**	**Between Group (*p* value) ^5^**
***Physical activity** (MET hours/day)*									
Sedentary activities	8.9 ± 4.7	8.6 ± 4.6	−0.3 (−1.0, 0.4)	0.6	8.4 ± 4.7	8.2 ± 4.8	−0.2 (−1.0, 0.7)	0.72	0.77
Moderate intensity	3.7 ± 5.6	2.6 ± 4.7	−1.1 (−2.3, 0.2)	0.1	3.9 ± 5.4	3.0 ± 5.4	−0.9 (−2.0, 0.2)	**0.049**	0.78
Vigorous intensity	1.1 ± 4.5	0.2 ± 0.5	−1.0 (−2.2, 0.3)	0.33	0.3 ± 1.6	0.1 ± 0.5	−0.2 (−0.6, 0.3)	0.89	0.63
***Weight** (kg)*	87.2 ± 19.1	87.0 ± 19.1	−0.15 (−0.7, 0.4)	0.56	88.0 ± 20.1	88.2 ± 20.4	0.20 (−0.22, 6.2)	0.35	0.29

MET – metabolic equivalent of task. Data are presented as mean ± SD or medians and interquartile ranges (IQR). ^1^ There were no significant between-group differences in baseline values; ^2^ mean change was calculated as week 12- week 0 and is presented as mean change (95% CI); ^3^ median change was calculated as week 12- week 0 and is presented as median change (IQR); ^4^ within-group comparisons were performed using paired sample t tests and Wilcoxon signed-rank test (*p* < 0.05); ^5^ between-group comparisons were made using independent sample t-tests and Mann–Whitney U test (*p* < 0.05); ^6^ all variables are logarithmically transformed and summarised as geometric medians (IQ range) and change as geometric median change (IQR).

**Table 3 nutrients-12-02445-t003:** Changes in mood, self-esteem, body image and quality of life indicators according to the Polyphenol Intervention Trial (PPhIT) study group allocation.

	Low Polyphenol Diet (*n*_max_ = 50)				High Polyphenol Diet (*n*_max_ = 49)				
	Week 0 ^1^	Week 12	Median Change (IQR) ^2^	Within Group Change (*p* Value) ^3^	Week 0 ^1^	Week 12	Median Change (IQR) ^2^	Within Group Change (*p* Value) ^3^	Between Group Change (*p* Value) ^4^
**BDI-II ***	6.0 (2.0, 12.5)	7.0 (2.0, 11.0)	0.2 (−1.5, 1.9)	0.98	6.0 (3.0, 12.0)	2.0 (0.0, 6.0)	−3.4 (−5.4, −1.5)	**<0.001**	**0.01**
**DASS-21 ****									
Depression ^5^	2.0 (0.0, 12.0)	6.0 (0.0, 10.5)	0 (−2.0, 6.0)	0.29	2.0 (0.0, 10.0)	0.0 (0.0, 6.0)	0 (−2.0, 0.0)	0.53	0.56
Anxiety ^5^	4.0 (2.0, 9.0)	2.0 (0.0, 8.0)	0.0 (−3.0, 3.0)	0.86	4.0 (0.0, 10.0)	2.0 (0.0, 6.0)	0.0 (−2.0, 0.0)	0.16	0.8
Stress ^5^	7.0 (2.0, 12.5)	8.0 (0.0, 16.0)	0 (−2.0, 4.0)	0.76	6.0 (2.0, 14.0)	4.0 (0.0, 10.0)	−2.0 (−6.0, 2.0)	**0.05**	0.14
**PANAS *****									
Positive affect	29.9 (8.3)	30.4 (9.8)	0.5 (−1.5, 2.5)	0.63	33.0 (6.8)	35.2 (7.4)	2.2 (0.3, 4.1)	**0.03**	0.21
Negative affect	11.0 (10.0, 13.0)	11.0 (10.0, 13.0)	0.0 (−1.5, 1.5)	0.56	12.0 (10.0, 15.0)	10.0 (10.0, 14.0)	0.0 (−1.0, 0.5)	0.68	0.99
**Rosenberg Self-Esteem Score ** ^**†**^	26.0 (25.0, 28.0)	26.0 (25.0, 28.0)	0.0 (−2.0, 2.0)	0.74	26.0 (25.0, 27.0)	27.0 (24.0, 27.0)	0.0 (−2.0, 2.0)	0.68	0.53
**MBSRQ-AS ** ^**††**^									
Appearance Evaluation	2.9 (2.4, 3.5)	3.0 (2.4, 3.6)	0.0 (−0.3, 0.4)	0.27	3.0 (2.5, 3.4)	3.1 (2.7, 3.6)	0.1 (−0.3, 0.4)	0.15	0.76
Appearance Orientation	3.0 (2.5, 3.7)	3.0 (2.5, 3.7)	0.0 (−0.3, 0.2)	0.35	3.2 (2.8, 3.5)	3.3 (2.9, 3.8)	0.2 (−0.2, 0.4)	0.16	0.1
Body areas Satisfaction	3.0 (2.7, 3.4)	3.3 (2.4, 3.7)	0.2 (−0.1, 0.4)	**0.03**	3.1 (2.7, 3.6)	3.3 (2.8, 3.8)	0.2 (−0.1, 0.3)	**0.02**	0.71
Overweight Preoccupation	2.3 (1.8, 2.8)	2.4 (1.8, 2.8)	0.0 (−0.3, 0.5)	0.87	2.5 (1.8, 3.3)	2.6 (1.8, 3.2)	0.0 (−0.5, 0.3)	0.45	0.72
Self-classified Weight ^1^	4.0 (3.4, 4.0)	4.0 (3.0, 4.0)	0.0 (0.0, 0.0)	0.08	4.0 (3.5, 4.0)	4.0 (3.5, 4.0)	0.0 (0.0, 0.0)	0.43	0.5
**SF-36 ^†††^**									
Physical Functioning	90 (75.0, 97.5)	90 (81.3, 100.0)	0 (0.0, 10.0)	0.07	95 (80.0, 100.0)	95 (85.0, 100.0)	0 (−5.0, 10.0)	0.15	0.44
Role limitations—physical health	100 (37.5, 100.0)	100 (37.5, 100.0)	0 (0.0, 25.0)	0.45	100 (75.0, 100.0)	100 (100.0, 100.0)	0 (0.0, 0.0)	**<0.001**	0.61
Pain	80 (47.5, 100.0)	80 (46.3, 100.0)	0 (−10.0, 10.0)	0.64	90 (60.0, 90.0)	90 (70.0, 100.0)	0 (−10.0, 22.5)	0.2	0.51
General health	65 (45.0, 75.0)	60 (50.0, 75.0)	0 (−10.0, 10.0)	0.47	65 (50.0, 75.0)	75 (65.0, 85.0)	10 (−5.0, 20.0)	**<0.001**	**0.03**
Physical health component	210.5 (168.5, 223.1)	200.9 (172.9, 217.2)	−6.4 (−17.0, 4.2)	0.09	213.2 (189.6, 225.1)	216.6 (201.4, 225.3)	2.2 (−8.1, 15.4)	0.2	**0.04**
Role limitations—emotional health	100 (100.0, 100.0)	100 (100.0, 100.0)	0 (0.0, 0.0)	0.99	100 (100.0, 100.0)	100 (100.0, 100.0)	0 (0.0, 0.0)	0.1	0.85
Energy/fatigue	55 (40.0, 72.5)	60 (45.0, 70.0)	5 (−5.0, 10.0)	0.39	60 (50.0, 80.0)	70 (60.0, 80.0)	5 (0.0, 20.0)	**<0.001**	**0.02**
Emotional well-being	76 (64.0, 84.0)	80 (62.0, 86.0)	0 (−8.0, 8.0)	0.73	80 (60.0, 88.0)	84 (72.0, 92.0)	4 (0.0, 16.0)	0	**0.01**
Social functioning	100 (75.0, 100.0)	100 (75.0, 100.0)	0 (−6.3, 0.0)	0.97	100 (75.0, 100.0)	100 (100.0, 100.0)	0 (0.0, 25.0)	**0.02**	0.08
Mental health component	209.1 (176.8, 222.3)	197.9 (175.0, 217.2)	−4.0 (−26.6, 8.1)	0.04	208 (181.4, 226.0)	218.3 (201.4, 226.8)	1.9 (−6.9, 19.1)	0.09	**0.01**

Data are presented as mean ± SD or medians and interquartile ranges (IQR). **^1^** There were no significant between-group differences in baseline values; ^2^ median change was calculated as week 12- week 0 and is presented as median change (IQR); ^3^ within-group comparisons were performed using paired sample t test or Wilcoxon signed-rank test (*p* < 0.05); ^4^ between-group comparisons were made using Mann–Whitney U test for continuous variables (*p* < 0.05); ^5^
*n* = 57 (LPD (*n* = 27), HPD (*n* = 30)); * BDI-II; Beck Depression Inventory Second Edition. Scores 0–13 = minimal depression, 14–19 = mild depression, 20–28 = moderate depression, 29–63 = severe depression; ** DASS-21; Depression Anxiety and Stress Scale 21 items. Depression score 0–9 = normal, Anxiety score 0–7 = normal, Stress score 0–14 = normal; *** PANAS; Positive and Negative Affect Scale. Higher score indicates higher positive and negative affect; ^†^ Rosenberg Self-Esteem Score; scores range from 0 to 30. Higher scores are indicative of higher self-esteem; ^††^ MBSRQ-AS; Multi-Dimensional Body-Self Relations Questionnaire—Appearance Scales. Higher scores indicative of higher body image satisfaction; ^†††^ Mental and physical health assessed using the RAND 36-Item Short Form Survey (SF-36). Physical functioning scores: “low” = limited a lot in performing all physical activities including bathing or dressing, “high” = performs all types of physical activities including the most vigorous without limitations due to health; Role limitations due to physical problems: “low” = problems with work or other daily activities as a result of physical health, “high” = no problems with work or other daily activities as result of physical health, past 4 weeks; Pain: “high” = very severe and extremely limiting pain, “low” = no pain or limitations due to pain, past 4 weeks; General health perceptions: “high” = believes personal health is poor and likely to get worse, “low” = believes personal health is excellent. Physical health component = sum of physical functioning, role limitations—physical health, pain and general health. Role limitations due to emotional problems: “high” = problems with work or other daily activities as a result of emotional problems, “low” = no problems with work or other daily activities as result of emotional problems, past 4 weeks; Energy/fatigue: “low” = feels tired and worn out all of the time, “low” = feels full of pep and energy all of the time, past 4 weeks; Emotional well-being: “high” = feelings of nervousness and depression all of the time, “low” = feels peaceful, happy and calm all of the time, past 4 weeks; Social functioning: “low” = extreme and frequent interference with normal social activities due to physical and emotional problems, “high” = performs normal social activities without interference due to physical or emotional problems, past 4 weeks. Mental health component = sum of role limitations—emotional health, energy/fatigue, emotional well-being and social functioning.

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
