# Peer review of "A High Polyphenol Diet Improves Psychological Well-Being: The Polyphenol Intervention Trial (PPhIT)"

_nutrients, 2020, doi:10.3390/nu12082445_

Round 1
Reviewer 1 Report
This work describes the beneficial effects including reduction in depressed symptoms, improvements in general mental health status, and physical health by dietary intake of a high polyphenol diet in hypertensive participants. This manuscript is well written and provides interesting insights, but there are some queries to accept in this journal.
- In Table 2, the authors showed data of micronutrient biomarkers, but only epicatechin data was urinary concentration. Please clarify the information in the table legend.
- Table 2. Significant differences of total carotenoids, lutein and zeaxanthin were observed between Low and High polyphenol Diet at Week 0. For example, the total carotenoids of Low polyphenol diet at week 0 was 44.30, while that of High polyphenol diet was 1.17. Is it correct?
- The parenthesis in the tables is difficult to understand. For example, in the case of SD, mean ±SD is popular.
Reviewer 2 Report
Congratulations! It makes a few years that I don't see such important and interesting study.
Author Response
Thank you for the feedback.